# The Emergence of H7N7 Highly Pathogenic Avian Influenza Virus from Low Pathogenicity Avian Influenza Virus Using an *in ovo* Embryo Culture Model

**DOI:** 10.3390/v12090920

**Published:** 2020-08-21

**Authors:** Amanda H. Seekings, Wendy A. Howard, Alejandro Nuñéz, Marek J. Slomka, Ashley C. Banyard, Daniel Hicks, Richard J. Ellis, Javier Nuñéz-García, Lorian C. Hartgroves, Wendy S. Barclay, Jill Banks, Ian H. Brown

**Affiliations:** 1Virology Department, Animal and Plant Health Agency (APHA-Weybridge), Addlestone, Surrey KT15 3NB, UK; gwendyddanne@yahoo.co.uk (W.A.H.); marek.slomka@apha.gov.uk (M.J.S.); Ashley.Banyard@apha.gov.uk (A.C.B.); jillybanks@ntlworld.com (J.B.); Ian.Brown@apha.gov.uk (I.H.B.); 2Pathology Department, Animal and Plant Health Agency (APHA-Weybridge), Addlestone, Surrey KT15 3NB, UK; alejandro.nunez@apha.gov.uk (A.N.); Daniel.Hicks@apha.gov.uk (D.H.); 3School of Life Sciences, University of Sussex, Falmer, Brighton BN1 9QG, UK; 4Institute for Infection and Immunity, St. George’s Hospital Medical School, University of London, London SW17 0RE, UK; 5Surveillance and Laboratory Services Department, Animal and Plant Health Agency (APHA-Weybridge), Addlestone, Surrey KT15 3NB, UK; Richard.Ellis@apha.gov.uk (R.J.E.); javier.nunez@apha.gov.uk (J.N.-G.); 6Virology Department, Imperial College, London W2 1NY, UK; lorianh@gmail.com (L.C.H.); w.barclay@imperial.ac.uk (W.S.B.)

**Keywords:** highly pathogenic avian influenza (HPAI), low pathogenicity avian influenza (LPAI), H7N7, reverse genetics, embryo culture, deep sequencing, immunohistochemistry

## Abstract

Outbreaks of highly pathogenic avian influenza virus (HPAIV) often result in the infection of millions of poultry, causing up to 100% mortality. HPAIV has been shown to emerge from low pathogenicity avian influenza virus (LPAIV) in field outbreaks. Direct evidence for the emergence of H7N7 HPAIV from a LPAIV precursor with a rare di-basic cleavage site (DBCS) was identified in the UK in 2008. The DBCS contained an additional basic amino acid compared to commonly circulating LPAIVs that harbor a single-basic amino acid at the cleavage site (SBCS). Using reverse genetics, outbreak HPAIVs were rescued with a DBCS (H7N7_DB_), as seen in the LPAIV precursor or an SBCS representative of common H7 LPAIVs (H7N7_SB_). Passage of H7N7_DB_ in chicken embryo tissues showed spontaneous evolution to a HPAIV. In contrast, deep sequencing of extracts from embryo tissues in which H7N7_SB_ was serially passaged showed retention of the LPAIV genotype. Thus, in chicken embryos, an H7N7 virus containing a DBCS appears naturally unstable, enabling rapid evolution to HPAIV. Evaluation in embryo tissue presents a useful approach to study AIV evolution and allows a laboratory-based dissection of molecular mechanisms behind the emergence of HPAIV.

## 1. Introduction

Avian influenza viruses (AIVs) are segmented negative sense RNA viruses belonging to the family Orthomyxoviridae [1]. They are divided into subtypes determined by the external viral glycoproteins, namely haemagglutinin (HA:H1-H16) and neuraminidase (NA: N1-N9), and normally occur as low pathogenicity (LP)AIVs, causing few (if any) clinical signs in wild birds and poultry [2,3]. The H5 and H7 subtypes may mutate to the highly pathogenic (HP)AIV variant typified by severe morbidity and mortality in domesticated galliforme species. The major genetic determinant of H5 and H7 pathogenicity occurs at the cleavage site (CS) in the HA protein [4,5]. Mutation to HPAIV is characterised by insertion of basic amino acids to produce a multi-basic cleavage site (MBCS) which is recognised by endogenous furin-like proteases, which enable HA cleavage and viral replication. These ubiquitous enzymes occur throughout multiple organ systems, resulting in systemic HPAIV infection. LPAIVs differ by possessing a single-basic cleavage site (SBCS) which requires exogenous trypsin-like proteases for HA cleavage and viral entry into cells for subsequent replication. These proteases are restricted to the respiratory and gastrointestinal tracts, thereby limiting LPAIV tissue tropism with usually very mild pathogenesis outcomes. The direct association of the MBCS with HA cleavability in vitro, and pathogenicity in vivo, has been demonstrated [6,7,8,9,10,11]. Critically, pathogenicity appears to be a polygenic trait and residues other than the MBCS contribute to an increase in virulence. Specifically, amino acids neighboring the CS in the HA protein and mutations in other viral proteins have been identified [12,13].

HPAIV outbreaks appear to originate from previous H5/H7 LPAIV incursions from wild birds to poultry, followed by mutation in galliformes (chickens or turkeys) to yield the corresponding HPAIV variant. Earlier LPAIV precursors to HPAIV outbreaks have been documented: Ontario Canada (H5N9) 1966 [14], Australia (H7N7) 1976 [15], Pennsylvania USA (H5N2) 1983 [16], Mexico (H5N2) 1994-1995 [17,18,19], Pakistan (H7N3) 1994-1995 [20], Italy (H7N1) 1999-2000 [21], Chile (H7N3) 2002 [22], Netherlands (H7N7) 2003 [23], Canada (H7N3) 2004 [24] and 2007 [25], England (H7N7) 2008 [26] and 2015 [27], Spain (H7N7) 2009 [28], Italy (H7N7) 2013 [29], Germany (H7N7) 2015 [30], Indiana USA (H7N8) 2016 [31], Italy (H7N7) 2016 [32] and Tennessee USA (H7N9) 2017 [33]. H7N9 LPAIV has circulated in China since March 2013 and continues to evolve [34], with eventual mutation to a corresponding HPAIV reported by February 2017 [35,36]. Mutation mechanisms at the HA CS have been proposed from past outbreaks [37], including the addition of basic amino acids due to polymerase slippage or stuttering during transcription [17,18], changes in the secondary structure in the CS region [38] and non-homologous recombination [21,22,24]. Understanding why some strains of H5 and H7 LPAIV can mutate relatively readily to become HPAIV and others do not remains a key unanswered question.

H7 outbreak investigations by reverse genetics (RG) and deep sequencing have addressed the insertion or composition of residues at the CS, analysing both the initial LPAIV sequences followed by HPAIV emergence and the distribution of subsequent HPAIV variants [11,39,40,41,42].

The prior existence of minority populations of more virulent viruses was demonstrated by plaque-purification from H5N2 and H7N2 LPAIV preparations using trypsin-free medium [43,44], with the latter containing an insertion of three basic residues at the CS and demonstrating a HPAIV phenotype in chickens. Multiple *in ovo* passages elicited the emergence of HPAIVs from H5N2 LPAIV precursors, which included CS sequence changes and increased virulence (with characteristic systemic organ tropism) in chickens, thereby demonstrating the value of this approach [45]. Viral subpopulations with differing pathogenic properties may be selected in the presence of adaptation pressure [46,47]. The generation of HPAIVs from H5N1 and H5N3 LPAIVs was achieved by passages through chicken air sacs followed by cerebral passage [6,48]. While these reports describe increasing virus pathogenicity through likely minority HPAIV selection during passaging experiments, none have recreated spontaneous HPAIV emergence from LPAIV by a single passaging method alone in the laboratory that represents the emergence of HPAIV in the field.

Differences in tissue tropism between LPAIVs and HPAIVs occur following *in ovo* inoculation of chicken-embryonated fowls’ eggs (EFEs). LPAIV replication in EFEs differs depending on the inoculation route, with allantoic cavity tropism facilitated by trypsin-like proteases, and viable LPAIV progeny are released into the allantoic fluid. Chorioallantoic membrane (CAM) inoculation restricts LPAIV replication to the site of introduction [49,50]. In contrast, HPAIV replication is not restricted and disseminates throughout the allantoic epithelium, mesenchyme and chorionic epithelium following allantoic or CAM inoculation. HPAIV infection can result in systemic spread within the embryo, with endothelial tropism apparent in the blood vessels, which also occurs in free-living chickens [50,51,52]. Investigation of H7N1 HPAIV in chicken, turkey, Muscovy duck and mallard embryos showed that all species supported HPAIV replication with virus spread observed in endothelial cells of the CAM and embryo tissues. H7N1 LPAIVs were restricted to the epithelial cells of the CAM in all embryo species when inoculated allantoically, but surprisingly, LPAIV spread to the skin, oesophagus and respiratory tract of mallard embryos [53].

H7N7 HPAIV caused an outbreak in 2008 in a layer farm in the UK [26], with three distinct HPAIV CS motifs (PEIPKRKKR/GLF, PEIPKKKKR/GLF and PEIPKKKKKKR/GLF) identified. Clinical, epidemiological and virological investigations provided evidence for an initial LPAIV incursion and circulation on the premises followed by mutation to HPAIV. Molecular analyses of samples collected from the environment revealed a viral subpopulation with a rare di-basic CS (DBCS) motif (PEIPKKR/GLF). This DBCS motif was also found in an H7N7 LPAIV from a mallard from Sweden in 2008, demonstrating its natural occurrence [54]. While a viable LPAIV was not isolated from the UK outbreak, statutory intravenous pathogenicity index (IVPI) testing of the Swedish isolate resulted in an IVPI of 0.00, confirming a LPAIV phenotype. Sequence comparisons of the outbreak virus and the Swedish LPAIV revealed high similarity in the HA gene only, with other segments being phylogenetically divergent [26]. Previously, this DBCS was identified in an LPAIV from ducks (Australia, 1976) which subsequently mutated to HPAIV after transmission to chickens [15,55]. Therefore, as the LPAIV isolates contained a DBCS instead of the typical SBCS, but remained as a LPAIV in vivo, it could be considered as an intermediate between LPAIV and HPAIV involved in the transmission chain from wild aquatic birds to domestic poultry.

Discovery of the DBCS at the UK H7N7 (2008) HPAIV outbreak prompted the current investigation of the emergence of HPAIV from an LPAIV precursor. AIV outbreaks in commercial poultry flocks are clearly characterised by an extremely large number of individual infections and transmission events, the scale of which would be impractical and prohibitive to reproduce in an in vivo experimental setting, and may also exclude other field factors (e.g., environmental) which may be conducive to the emergence of HPAIV. As noted above, previous studies have demonstrated the emergence of minority viral populations, which include the detection of increased proportions of previously existing HPAIV variants. Importantly, the current study differs in that it is not reliant on such HPAIV emergence from any pre-existing viral sub-population. Instead, the novelty of the current study exploits RG technology to provide clonally generated RG-viruses passaged in an *in ovo* embryo culture model to drive the emergence of HPAIV. Should HPAIV emergence then occur, such an event may be judged to be a more spontaneous LPAIV-to-HPAIV switch, as opposed to the amplification of a pre-existing minority HPAIV population within the inoculum. Assessing the potential of an LPAIV to mutate into HPAIV once introduced into galliforme poultry is required to understand why some LPAIV strains rapidly mutate to become HPAIV, while others do not.

## 2. Materials and Methods

### 2.1. Ethics and Biosafety Statement

Animal experiments were approved by the APHA Animal Welfare and Ethical Review Body (70/8332-9-002, 20/5/2015) in accord with UK and European legislation. Laboratory and animal experiments were conducted in UK approved SAPO level 4, ACDP level 3 biocontainment facilities at APHA-Weybridge, UK [56].

### 2.2. Cells

Primary chicken embryo fibroblasts (CEFs) and cell lines of Madin-Darby canine kidney (MDCK) and human embryonic kidney (HEK 293T) origin were maintained in Dulbecco’s modified Eagle’s medium (DMEM) supplemented with 10% foetal calf serum and 1% penicillin-streptomycin (Invitrogen, Renfrew, UK). Incubation of all cells was at 37 °C in a 5% CO_2_ atmosphere.

### 2.3. Viruses

The eight H7N7 viruses used in this study are listed in Table 1. A 12-plasmid RG system was used to rescue recombinant viruses [57], comprising eight rescue plasmids (pPol I) based on A/chicken/England/11406/2008 H7N7 (England-08; GISAID accession # EPI712884-EPI712891 [26]) with four helper plasmids (pPol II) that express the viral polymerase genes and nucleoprotein from A/Victoria/3/75 (H3N2) [58]. Plasmids were synthesised by GeneArt^TM^ Gene Synthesis (ThermoFisher Scientific, Paisley, UK). Modifications to the HA pPol I plasmid were performed using QuickChange II XL Site-Directed Mutagenesis Kit (primers available on request) following the manufacturer’s instructions (Agilent Technologies, Stockport, UK). All 12 plasmids were transfected into 293T cells in 12-well plates using FUGENE HD transfection reagent (Promega, Southampton, UK). After overnight incubation, cells were removed and co-cultured with MDCK cells in 25 cm^2^ flasks. After 6–8 h incubation the cells were washed and 5 mL fresh serum-free DMEM was added. Where appropriate, TPCK trypsin was added at 0.5 μg/mL (Fisher Scientific, UK). Rescued “synthetic” viruses were passaged once in 9-11–day-old EFEs to increase their titer and also to obtain an avian origin envelope as per the wildtype (wt) viruses. Viruses were full-genome sequenced to confirm the introduced genetic changes.

### 2.4. Titration According to Plaque-Forming Units (pfu)

Infectious viral titres were determined by plaque assays using 100% confluent MDCK cells. Ten-fold dilution series of viruses were added to wells in duplicate and overlay DMEM media containing 2.5% agar (Oxoid, Basingstoke, UK). Where appropriate, TPCK trypsin was included at 2 μg/mL (Fisher Scientific, UK). After 72 h incubation, cells were stained with 1% crystal violet (Sigma, Poole, UK) and visible plaques counted. Viruses were titrated in triplicate and titres were expressed as pfu/mL.

### 2.5. Multi-Cycle Growth Kinetics

Titrated viruses infected confluent MDCK cells at a multiplicity of infection (MOI) of 0.0001. Where appropriate, TPCK trypsin was added at 0.5 μg/mL. Supernatants were removed from each well at 8, 24, 48, and 72 h post infection (hpi).

### 2.6. Viral RNA Isolation

Viral RNA was extracted manually using the QIAquick Viral RNA extraction kit (QIAGEN, Manchester, UK) [59]. RNA was extracted from formalin-fixed paraffin-embedded (FFPE) embryonic tissues using RecoverAll™ Total Nucleic Acid Isolation Kit for FFPE (Ambion, Paisley, UK) according to the manufacturer’s instructions.

### 2.7. M-Gene Reverse Transcription RealTime (RRT)-PCR

M-gene RRT-PCR was carried out using primers and probe [60] as described [59]. Viral RNA was quantified as relative equivalent units (REUs) against a 10-fold dilution series of EID_50_ quantified RNA [61].

### 2.8. Intravenous Pathogenicity Index (IVPI) Determination

Standard IVPI tests were carried out [2], with each SPF chicken inoculated intravenously with 0.1 mL allantoic fluid diluted to a specified titer or a haemagglutinating activity titer >2^4^. Welfare considerations and IVPI scoring were applied as described [56], with an IVPI of ≥1.2 considered as an HPAIV phenotype [2].

### 2.9. Embryonated Fowls’ Eggs (EFEs)

#### 2.9.1. Virus Isolation and Propagation in EFEs

Nine- to eleven-day-old specific pathogen-free (SPF) EFEs were used for virus isolation, growth and titration of median egg infectious dose (EID_50_/mL) by standard means [2,62].

#### 2.9.2. Allantoic Fluid Passage Experiments of H7N7_DB_

H7N7_DB_ RG virus was blind passaged ten times in 14-day-old (14do) EFEs. Thirty eggs were inoculated via the allantoic cavity with 0.1 mL per egg containing 10^5^ EID_50_ H7N7_DB_ virus. Following embryo death, or after 72 hpi, 1 mL allantoic fluid was harvested from each egg and pooled. Subsequent passages were carried out in two groups each consisting of 15 14do EFEs inoculated with either 0.1 mL neat allantoic fluid from the previous passage (neat group) or this material diluted 1:5000 in PBS (diluted group). At each passage, egg death was recorded and HA activity of the allantoic fluid was determined [2].

#### 2.9.3. Tissue Tropism in EFEs

Nine-day-old (9do) and 14do EFEs were each inoculated with 0.1 mL of either 10^1^ EID_50_ or 10^4^ EID_50_ virus. Embryos were harvested after 37 °C incubation at 24, 48, and 72 hpi. The allantoic fluid and membranes (allantoic and amniotic) were harvested and the membranes fixed in 10% buffered formalin. Each embryo was removed from the egg, humanely killed by disruption of membranes and decapitation (cessation of circulation), embryo head and body tissues were fixed in the same pot of 10% buffered formalin. The method was later refined to include only 14do embryos inoculated with 0.1 mL 10^1^ EID_50_ and harvested at 72 hpi. Embryos were bisected longitudinally; one half of the head and body was fixed in 10% buffered formalin for IHC as described [61] and the other remained unfixed for subsequent homogenization for further characterisation, which included EFE passage, RNA extraction, DNA Sanger or deep sequencing.

### 2.10. Sanger Sequencing

DNA sequencing using the BigDye v3.1 kit (Applied Biosystems, Warrington, UK) used gene-specific primers (available on request), with analysis on an ABI Prism 3130 Genetic Analyser (Applied Biosystems, Warrington, UK). The Lasergene package version 12 (DNAStar, USA) was used for nucleotide sequence analysis.

### 2.11. Deep Sequencing

Deep sequencing was carried out on the MiSeq Illumina platform directly from extracted viral RNA or PCR-amplified across the CS region (see Appendix A). DNA library preparation was performed using the Illumina Nextera^®^ XT Library Prep Kit (Illumina, Cambridge, UK) according to the manufacturer’s instructions. Indexed libraries were quantified and pooled in equimolar concentrations (2 nM final concentration) and sequenced in multiplex with 2 × 150 base-paired end reads on the MiSeq Illumina platform. Adaptors and primers were removed and reads with a Phred quality score below 30 were excluded using Trimmomatic [63]. Reads shorter than 120bp were also excluded. Raw sequence reads were mapped to the genome of A/chicken/England/11406/2008 H7N7 (accession numbers EPI712884-EPI712891) England-08 using a BWA-MEM algorithm [64]. The consensus sequence was extracted from the resultant bam file using a modified SAMtools script [65] (vcf2consensus.pl available at: https://github.com/ellisrichardj/csu_scripts/blob/master/vcf2consensus.pl). The frequency of background variants generated by experimental and technical error were established (see Appendix A) and the derived error level, 1.92%, was applied to sequenced samples.

### 2.12. Statistics

Statistical analysis for multi-cycle growth kinetics were performed using GraphPad Prism version 6.04. Individual Log values were transformed using the equation Y=Log(Y). The statistical significance of virus growth was assessed by a two-way ANOVA with Bonferroni post-test correction for multiple comparisons. The binomial test of proportions was used to calculate the sample size required to estimate a prevalence of 0% with a 97.5% one-sided confidence interval of 0.05–0% of infected embryos (*n* = 72).

## 3. Results

### 3.1. Rescue of Recombinant RG Viruses

An RG-copy of wt England-08 containing a MBCS was successfully rescued along with RG viruses containing the DBCS from the LPAIV precursor to the HPAIV outbreak (H7N7_DB_) and a typical SBCS of circulating viruses in Eurasia (H7N7_SB_). Previous isolation of wt England-08 in EFEs resulted in sequence changes in the HA gene compared to the sequence obtained from the original clinical material. Consequently, the three different CS motifs were rescued on both H7N7 HA backgrounds (Table 1). Full genome sequencing of each of the rescued recombinant viruses revealed no other genetic changes in all eight genome segments.

### 3.2. A MBCS Reflects a Trypsin-Independent Phenotype In Vitro

H7N7_SB_ and H7N7_DB_ required trypsin to produce plaques (Figure 1a) and grew to lower titres in MDCK cells than H7N7_MB_ in the absence of trypsin at 24 hpi (*p* < 0.001), 48 hpi and 72 hpi (*p* < 0.0001) (Figure 1b). In contrast, monolayers infected with H7N7_MB_ produced plaques without trypsin (Figure 1a) with growth independent of trypsin (Figure 1b). H7N7_MB_ achieved a slightly higher, although not statistically significant, titre in the absence of trypsin.

### 3.3. A MBCS in the HA Protein Is Required for a HP Phenotype In Vivo

Five groups of chickens were inoculated as per the IVPI test. IVPI chickens were previously infected with England-08 [26] and are included to compare pathogenicity of the various H7N7 viruses (Figure 2). Following infection with England-08 and H7N7_MB+_, 100% mortality occurred, although the former all died very rapidly by 1 dpi. While the H7N7_MB_ attained 50% mortality, all three registered IVPI scores of >1.2 as evidence of being HPAIVs (Figure 2). Following the observation that H7N7_MB_+ gave a higher IVPI than H7N7_MB_, the H7N7_DB_+ and H7N7_SB_+ rescued viruses were then selected for IVPI testing. Neither virus showed clinical signs throughout the duration, with both registering IVPI values of 0.00, indicating an LPAIV phenotype (Figure 2).

### 3.4. H7N7_DB_ Passage in Allantoic Fluid

The effect of *in ovo* passage of H7N7_DB_ was initially assessed by sequential passage of allantoic fluid in 14do EFEs. No substantial change in embryo death time or allantoic fluid HA titres (all >1:64) across 10 passages were seen. Sanger sequencing of passage 10 material did not reveal any genetic changes at the consensus level, with the HA sequence being identical to the initial inoculum. Four non-synonymous changes were observed outside of the CS region (S31I, F92V, N123D, G196E; H7 numbering of the mature protein) but with no evidence of any MBCS emergence via *in ovo* passage.

### 3.5. Comparative Pathotyping of H7N7 Wt and Recombinant RG Viruses in Chicken Embryo Tissues

Nine-day-old and 14do EFEs were inoculated with two titres, 10^1^ EID_50_ and 10^4^ EID_50_, of the HPAIVs wt England-08 and RG H7N7_MB_ as well as the LPAIV wt Sweden-08 virus. Three embryos per titre were harvested after 37 °C incubation at 24, 48 and 72 hpi. Virus-specific IHC revealed differences between the HPAIVs and LPAIV which reflected pathotype distinction. In addition to viral tropism in the allantoic and amniotic membranes, both HPAIVs displayed systemic multi-organ distribution, including parenchymal and endothelial cells which defined an HP phenotype (Figure 3A,B). Widespread endothelial staining was prominent in lung, heart, spleen, liver and kidney tissues. The viral distribution in embryos for both the HPAIVs were comparable and no marked differences in antigen distribution were seen between the 9do and 14do EFEs infected with either dose. In contrast to the HPAIV infections, embryos infected with Sweden-08 showed dominant replication in the allantoic and amniotic epithelia from 24 hpi, independent of dose or age (shown as duplicate embryo findings; Figure 3C,D). These observations are consistent with typical LPAIV *in ovo* infection and are defined as an LP phenotype. Occasional dissemination of Sweden-08 was observed in some embryonic organs comparable to HPAIV-infected embryos e.g., in lung tissue (Figure 3C). Absence of endothelial tropism was another overall outcome of LPAIV infection, although staining of discrete foci of hepatocytes and endothelial cells in the liver of one embryo was detected (Figure 3D), indicative of a mixed LP/HP phenotype. The above IHC results obtained from *in ovo* infection with the two wt viruses (one LPAIV and one HPAIV) and the one recombinant HPAIV (Figure 3), in young and older aged embryos at two different doses, enabled clear distinction of three phenotypes: LP, HP and a mixed LP/HP phenotype.

### 3.6. Emergence of HPAIV after H7N7_DB_ Passage in Chicken Embryo Tissues

Following the identification of the mixed LP/HP phenotype in embryo tissues infected with Sweden-08, the role of the DBCS in the emergence of HPAIV was investigated. Two different titers of H7N7_DB_ (10^1^ EID_50_ or 10^4^ EID_50_) were inoculated in 9do and 14do EFEs and harvested at 37 °C for 24, 48 and 72 hpi (*n* = 36 total). The majority of infected embryos showed characteristic LP phenotypes by IHC (Figure 4A). At 72 hpi, two embryos displayed a mixed LP/HP phenotype: one inoculated at 9do with 10^4^ EID_50_ (H7N7_DB_ 9do) and another inoculated at 14do with 10^1^ EID_50_ (H7N7_DB_ 14do). Positive cells were stained in the membranes of the mesenchyme and blood vessels with widespread labelling in blood vessel endothelium. The H7N7_DB_ 9do embryo had widespread multifocal labelling of endothelial cells and labelling in tubular epithelial cells. Moderately positive amnions in addition to positive sloughed cells in the parabronchi, hepatocytes and numerous splenic cells were observed. Positive labelling was also found in the brain, heart, kidney, intestine and bursa tissues (Figure 4B). The H7N7_DB_ 14do embryo also showed positive labelling in skeletal muscle cells, brain tissue, bursal follicles, spleen and hepatocytes in addition to amniotic cells (Figure 4C). To attempt reproducibility of the above mixed LP/HP phenotype findings, a repeat inoculation of nineteen 14do EFEs was done with 10^1^ EID_50_ of H7N7_DB_ and harvested at 72 hpi. Six of nineteen (32%) embryos displayed a mixed LP/HP phenotype by IHC.

RNA extracted from FFPE tissue or tissue homogenates from both experiments was Sanger-sequenced across the CS. Sequence was obtained from one 9do embryo and five 14do embryos that showed a mixed LP/HP phenotype at 72hpi and demonstrated that the H7N7_DB_ virus had acquired additional basic amino acids to create MBCS sequence motifs. Two different MBCS were sequenced: PEIPKKKKR/GLF and PEIPKRKKR/GLF, both of which occurred naturally during the 2008 H7N7 UK outbreak [26]. RNA was extracted from total tissue homogenates for deep sequencing from amplicons. Concordant with Sanger sequencing, tissue homogenates from two embryos with a mixed LP/HP phenotype (H7N7_DB_ E3 and E10) contained a majority MBCS population at 53.10% and 69.24% respectively by deep sequencing (Figure 5 and Appendix A). However, deep sequencing revealed that a high proportion of DBCS also remained in the tissues at 43.50% and 29.59%, respectively (Figure 5 and Appendix A).

Interestingly, for another embryo (H7N7_DB_ E14), only the FFPE block from the body section displayed a mixed LP/HP phenotype (Figure 4D) and MBCS by Sanger sequencing. The head and membrane tissues had very few positively stained endothelial cells (Figure 4D) compared to the same sections in other embryos (Figure 4B,C). DNA Sanger sequencing revealed retention of the DBCS. Deep sequencing showed that the head and membrane FFPE tissues contained a majority DBCS population as expected (72.77%) but also enabled detection of MBCS variants, albeit at a lower proportion of the total variant (26.52% MBCS population; Figure 6 and Appendix A). The body section (visceral organs) contained a majority MBCS population at 98.94%, which was in accord with Sanger sequencing, with only 0.02% DBCS detected (i.e., below the error threshold of 1.92%; Figure 6 and Appendix A).

Despite the mixed LP/HP phenotype and emergence of MBCS in the embryo tissues, the corresponding allantoic fluids collected from six of the nineteen (32%) embryos inoculated with H7N7_DB_ in the repeat experiment still retained the DBCS and showed no evidence of mutation to MBCS by Sanger sequencing. Deep sequencing of the allantoic fluid for H7N7_DB_ E3 and H7N7_DB_ E10 also showed retention of the DBCS (Appendix A).

### 3.7. Confirming the Phenotype of the HP Progeny Viruses by IVPI

Virus propagation of H7N7_DB_ E3 and H7N7_DB_ E10 tissue homogenates that contained a majority population of MBCS was attempted in EFEs, however the harvested allantoic fluids contained DBCSs by Sanger sequencing. A further attempt to isolate a MBCS containing virus from the tissue homogenate was performed by enriching for MBCS populations in CEFs without trypsin prior to inoculation of EFEs. Sanger sequencing revealed the presence of an MBCS from the CEF supernatants inoculated with both tissue homogenates. Upon one further passage in EFEs, the allantoic fluid from H7N7_DB_ E10 successfully produced the MBCS: PEIPKKKKR/GLF, but allantoic fluid from H7N7_DB_ E3 retained the DBCS. By deep sequencing, the population of MBCS in the cell supernatant of H7N7_DB_ E3 slightly increased from 53.10% to 59.51%, but when the cell supernatant was inoculated into EFEs and the allantoic fluid harvested, the population of MBCS decreased to 32.28% (Figure 5 and Appendix A). The DBCS population appeared to have been preferentially selected and amplified to constitute the majority population at 63.99%. In contrast, the population of MBCS in the H7N7_DB_ E10 sample greatly increased from 69.24% in the tissue homogenate to 97.64% in the CEF supernatant. When inoculated into EFEs this only slightly decreased to 94.59% (Figure 5 and Appendix A).

The mixed genotype progeny virus from H7N7_DB_ E10, named H7N7_DB-MB_ hereafter, was tested by IVPI. The final IVPI result was 1.31 and confirms that H7N7_DB-MB_ is a HPAIV in chickens with a greater IVPI than the H7N7_MB_ (IVPI 1.26) (Figure 2). Mortality of chickens infected with H7N7_DB-MB_ was slower than H7N7_MB_, however, 60% (*n* = 6/10) of chickens infected with H7N7_DB-MB_ died compared to 50% (*n* = 5/10) infected with H7N7_MB_.

### 3.8. Generation of HPAIV from H7N7_SB_ Was Unsuccessful after Multiple Passages in Chicken Embryos

Seventy-two 14do EFEs were inoculated with 10^1^ EID_50_ H7N7_SB_, harvested at 72 hpi, and were classified as an LP phenotype (Figure 7A,B). However, 18 (25%) showed enhanced labelling in the amniotic epithelium, skin, lung, liver, intestine and bursa epithelium (Figure 7B). Four of the 18 embryos (22%) that showed the most prominent staining in these organs showed retention of the SBCS by both Sanger and deep sequencing (Figure 8 and Appendix A). The enhanced level of replication in systemic tissues warranted further investigation by an additional embryo passage. Reserved frozen embryo tissue from one of these embryos (P1: E14) was re-passaged into fourteen 14do EFEs. Two of the 14 embryos (14%) showed a mixed LP/HP phenotype by IHC (Figure 7C), essentially as defined previously. RNA was extracted from both the fresh tissue homogenate embryo half and the FFPE embryo tissue half. Sanger and deep sequencing from both embryo halves showed retention of the SBCS (P2: E14, E1 and P2: E14, E11) (Figure 8 and Appendix A).

Virus-specific staining in the brain without evidence of a CS mutation prompted one further passage (P3) to determine whether this genotype could be selected. Tissue homogenates from both embryos that showed a mixed LP/HP phenotype at passage two were passaged in a further fifteen EFEs. At P3, only one of 15 embryos (~7%) (P3: E14, E11, E9) showed a mixed LP/HP phenotype (Figure 7D). Epithelial and endothelial staining was observed in the amniotic membrane and the skin. A low number of positively stained neurons in the brain, spleen, liver and kidney were observed, but an SBCS genotype was detected by Sanger and deep sequencing with no evidence of mutation to an MBCS (Figure 8 and Appendix A). Full genome consensus sequencing was performed for three SBCS passaged tissue homogenates that demonstrated a mixed LP/HP phenotype by IHC (P2: E14, E1; P2: E14, E11; P3: E14, E11, E9). Non-synonymous mutations resulted in five and eight amino acid changes at P2 and P3 respectively, but the SBCS remained unchanged (Table 2).

## 4. Discussion

An H7N7 HPAIV RG-system was developed to successfully introduce mutations at the CS and in the HA gene to attempt to study HPAIV emergence. H7N7 viruses containing an MBCS replicated in vitro in the absence of trypsin, but DBCS- and SBCS-containing viruses required trypsin. There is very limited prior literature describing H7 viruses possessing a DBCS, and whether an extra basic amino acid at the CS contributes to enhanced HA cleavage resulting in increased pathogenicity for poultry. In vitro growth curve assays, measured using molecular tools, demonstrated low-level growth of H7N7_DB_ and H7N7_SB_ without trypsin (Figure 1b). Accumulation of progeny viral RNA over time suggested higher viral loads for H7N7_DB_ than for H7N7_SB_ in vitro in the absence of trypsin, although this was not statistically significant. Replication of LPAIVs in the absence of trypsin has been observed previously [66,67], with speculation that egg-cultured LPAIV can undergo subsequent replication in vitro in trypsin-free medium due to carry-over of proteases from the egg fluid [68]. Both H7N7_SB_ and H7N7_DB_ displayed an LPAIV phenotype and remained trypsin-dependent for plaque formation and optimal growth. Importantly, both viruses were not pathogenic for chickens (IVPI 0.00).

The successful spontaneous mutation of the DBCS to a MBCS via *in ovo* passage in this study suggested that this DBCS may represent an intermediate that is already “primed” to mutate further, allowing for basic amino acid residue accumulation. The rare detection of wt viruses containing the DBCS previously described only in waterfowl on two occasions [54,55] suggests that it is unstable or unfavorable for maintenance in galliforme poultry. Mutation to an MBCS from a DBCS occurs rapidly following transmission to chickens and is readily selected as the dominant viral population. Further work to evaluate mutation from LPAIV to HPAIV in different avian species is merited to explore the hypothesis that DBCS emergence may be restricted to waterfowl as opposed to gallinaceous poultry. However, whilst the DBCS may represent an intermediate, passaging of the DBCS virus resulted in an increase in the minority population of SBCS in one embryo (1.25% to 1.41%; Figure 5A) and whether this is a true result or an increase in sequencing error remains to be determined.

Examining the nucleotide composition encoding the two MBCS motifs, it is proposed that PEIPKKR/GLF mutated to PEIPKKKKR/GLF first by a duplication insertion event and then mutated to PEIPKRKKR/GLF by base substitution, although mutation to PEIPKRKKR/GLF directly from PEIPKKR/GLF cannot be ruled out. Further investigation of the molecular mechanisms involved in the stepwise mutation of the CS is required. The minimum number of basic amino acids required to result in an HPAIV phenotype have been investigated [42,69,70]. However, identifying the optimal nucleotide composition that allows for emergence of an HPAIV phenotype could improve our understanding on how basic amino acids accumulate at the CS in nature. Evidence from the H7N7 UK outbreak in 2008 was also unclear on which motif occurred first, although PEIPKKKKR/GLF was detected more often (*n* = 35) than PEIPKRKKR/GLF (*n* = 10) by consensus sequencing [26]. This observation was comparable to the *in ovo* passage where mutation to PEIPKKKKR/GLF was more frequent (*n* = 5) than PEIPKRKKR/GLF (*n* = 1). However, the selection and mutational pressures that occurred at the outbreak (29-week-old layers) are different to those in an experimental *in ovo* passage with RG viruses. Whilst it is accepted that live birds, rather than embryos, are key in the selection pathway from LPAIV to HPAIV, the *in ovo* embryo culture approach served as an alternative and reduced in vivo experimentation.

Interestingly, embryo age did not appear to influence the ability of H7N7_DB_ to mutate as this was seen in both 9do and 14do EFEs. This contrasted with earlier studies where an H5N2 virus acquired HPAIV status after passage in 14do but not in 8-10-day-old EFEs [45,71]. An increase in *in ovo* pathogenicity was observed when H7N7_DB_ was passaged in the EFE model, however it is imperative to determine whether this can also occur in hatched birds. Passage of H7N7_DB_ in chickens would be an essential experiment to validate this *in ovo* model as a substitute for in vivo experiments such as IVPI. This could contribute to the principles of replacement, reduction, and refinement of animals in scientific research [72] and provide supporting evidence to refine international approaches for defining highly pathogenic status [2].

Although the DBCS successfully mutated to a MBCS in the embryo tissue, it remains unclear why no virus containing an MBCS was detected in the allantoic fluid. Feldmann et al. (2000) described receptor expression and polar budding as factors that can restrict the spread of HPAIV from endothelial cells in embryo tissues. There is also the possibility that epithelial cells in the allantoic and amniotic cavities have already been infected with LPAIV prior to the emergence of HPAIV such that the number of susceptible cells is reduced (due to cell death), thereby restricting HPAIV spread. This may explain why no mutation to MBCS was observed when H7N7_DB_ was passaged ten times via the allantoic cavity. This observation may reflect a unique tropism of the virus or it may indicate that ten sequential passages in EFEs are insufficient for mutated virus release into the allantoic fluid. Interestingly, previous studies have shown that HPAIV emergence can occur in allantoic fluid after few EFE passages [45]. Since only the allantoic fluid-derived virus was sequenced, it is plausible that MBCS mutants remained undetected within the embryo tissues. Possible MBCS emergence during passage in allantoic fluid cannot be excluded as the pooling of allantoic fluid at each passage in the initial study may have inadvertently diluted any MBCS mutants, preserving DBCS dominance within the viral population. Consensus sequence data from allantoic fluid at the 10th passage demonstrated 100% conservation with input virus across the HA CS. This was further supported through sequencing of the allantoic fluid of infected embryos that had a mixed LP/HP phenotype. Indeed, it was not until the MBCS virus population was enriched by selection in vitro without trypsin that the MBCS virus was propagated to recover an H7N7_DB-MB_ virus. It has been speculated that cellular protease availability may change during embryo development [73], and that LPAIVs may be degraded in the allantoic fluid of older embryos (14do) [45], thereby allowing HPAIV minority variants to dominate the population.

Influenza A viruses exist as a collection of closely related viral quasispecies, with reports that minority HP viral variants can be maintained in a predominantly LPAIV population [71,74]. A significant impeding impact on H7 HPAIV replication when co-infected with H7 LPAIV has been described *in ovo* and in vivo [75], suggesting that there may be a threshold at which a particular population is preferentially selected for (or outgrows) the other. However, results from the multi-cycle growth curve demonstrated that neither LPAIVs (H7N7_SB_; H7N7_DB_) nor HPAIV (H7N7_MB_) demonstrated an inherent replicative advantage in vitro when measuring nucleic acid over time (Figure 2). The emergence of an HPAIV from an LPAIV can be attributed to spontaneous CS mutation or the selection of minority MBCS variants in the viral population [44,45,46,47,48,75]. Indeed, both events are likely to occur in nature where the emergence of HPAIV from LPAIV precursors have been reported. The successful spontaneous DBCS-to-MBCS mutation demonstrated that the *in ovo* passage model can support the emergence of an MBCS and mimics the selection of an HPAIV as seen in the field.

Mutation to an MBCS was not achieved when H7N7_SB_ was passaged in EFEs. One of the proposed molecular mechanisms for the insertion of multiple basic amino acids is polymerase slippage at purine bases during viral replication to introduce additional purines that translate into basic amino acids [37]. This mechanism may have been involved in the mutation of DBCS to MBCS and is theoretically plausible with the SBCS, although there was no evidence of naturally occurring SBCS-containing viruses during the H7N7 outbreak in 2008 [26]. Furthermore, with the RG viruses, H7N7_SB_ did not change to DBCS or MBCS where >98% of the population retained the same SBCS (Figure 8 and Appendix A). Therefore, the recombinant H7N7_SB_ may require additional consecutive purine bases to enable polymerase slippage or further adaptation/compensatory mutations to mutate further. This observation is comparable to investigations of evolution of H5 HPAIV [42], where it was speculated that viruses containing short CS motifs (≤2 basic residues) are more stable and require intense selection pressure for a mutational change at the CS compared to viruses that contain mid-length CS motifs (3–4 basic residues). Indeed, residues other than the MBCS may also be attributed to an increase in pathogenicity. Recombinant H2, H4, H8 and H14 viruses with engineered MBCS on a H5N1 HPAIV background could support an HPAIV phenotype [76] in addition to MBCS engineered into H6 and H9 subtypes [67,77].

Full-genome sequencing of the tissue homogenates from embryos infected with H7N7_SB_ that showed a mixed LP/HP phenotype also had multiple substitutions that could be contributing to this phenotype in the PA, HA, NA and M proteins (Table 2). The function of these identified substitutions remains unknown for an avian host. The accumulation of mutations seen from P2 to P3 may suggest that H7N7_SB_ is unstable and is selecting mutations for increased fitness *in ovo*. One confounder identified with this embryo model was the unexplained dissemination of LPAIV in the brain cells, particularly in endothelial cells and neurons. Literature suggests that an MBCS is required for intracellular cleavage and replication where trypsin-like serine proteases are absent [10,78]. However, replication in endothelial cells was observed in brain tissue harvested from chickens from an H4N6 LPAIV outbreak [79]. A plausible hypothesis could be that initial infection with LPAIV altered the permeability of the allantoic and amniotic membrane allowing, the virus to replicate in the epithelial layer and disseminated to embryo structures bathed by the amniotic fluid such as the skin, the respiratory and digestive epithelia. The allantoic membrane epithelium extends into the embryonic coelom connected to the allantoic fluid [80], which may explain how LPAIVs which are unable to replicate in endothelial cells are transported through by circulating cells to other sites. The mixed LP/HP phenotypes observed during H7N7_DB_ or H7N7_SB_ infection without an MBCS genotype may be explained as spillage from replication sites via macrophages that have disseminated into other tissues, giving the impression of an artefactual extended antigen dissemination. It is unclear whether these LPAIVs are replicating or if they are circulating in an un-cleaved, non-infectious and immature form after one round of replication. Determining the exact mechanism by which LPAIVs can enter systemic organs remains to be investigated. Further work, using relevant ex vivo assays such as chicken tracheal epithelial cells [81] or chicken tracheal organ cultures [82], is warranted.

In contrast to the H7N7_DB_ and H7N7_SB_ viruses, the H7N7_MB_ virus was virulent in vivo (IVPI: >1.2), proving that this was an HPAIV according to standard international definitions [2]. Interestingly, different IVPI scores were obtained for H7N7_MB_ and H7N7_MB_+. It appears that the substitutions in the HA protein outside of the CS, I77V and A123T, the latter introducing a potential glycosylation, was sufficient to substantially increase the IVPI from 1.26 to 2.70. The A123T substitution has been previously characterized as a poultry-adaptation marker [83] and may be principally responsible for this virulence increase as described in earlier observations [83,84], although the underlying pathogenicity mechanism remains unclear. However, the H7N7 viruses assessed in this study are unique and do not contain a glycosylation site at N149, nor do they have a stalk deletion in the NA protein, which have been associated with increased poultry replication and pathogenicity [85]. Despite H7N7_DB_+ and H7N7_SB_+ containing the additional A123T adaptive marker for chickens, it was insufficient to cause clinical disease in IVPI-inoculated chickens. It appears that while this marker enhances pathogenicity when coupled with an MBCS, it is not the sole virulence determinant. Banks and Plowright (2003) analysed 85 H7 and 62 H5 protein sequences for the presence of potential glycosylation sites, but found no direct correlation between glycosylation near the receptor binding site (RBS) and increased pathogenicity. However, after intracerebral passage in one-day-old chickens, additional glycosylation sites emerged and correlated with increased pathogenicity [86]. This observation is analogous to the H7N7 2008 UK outbreak viruses where glycosylation at N123 was not naturally selected in hens, but was selected from a minor variant in a HPAIV sample after EFE passage [26]. This finding indicated that an additional glycosylation site near the RBS is not a prerequisite for HPAIV evolution from LPAIV precursors, but suggested that they may predispose AIVs to increased virulence.

Chickens which were IVPI-inoculated with the wt England-08 isolate derived from the outbreak all died by 1 dpi and registered the maximum IVPI of 3.00. By comparison, the genetically identical RG virus H7N7_MB_+ caused a more gradual time to death and gave an IVPI of 2.7. These results suggest that the RG virus was not as fit or virulent as wt England-08, although this difference is unlikely to be statistically significant. This has been noted previously with other RG viruses [87]. There is an indication that there is a requirement for quasi-species, a cloud of genetic variants, in the viral population to enhance virulence as described for other viruses [88,89,90].

While genetic pressures are believed to play a key role in the mutation from LPAIV to HPAIV, the immunological status of the host and external pressures such as the environment must also be considered [91,92]. Whether an intermediate between an LPAIV and an HPAIV is an obligatory requirement for stepwise mutation remains unknown. Whether this may begin to occur in wild birds or only in poultry is yet to be fully investigated. Surveillance of AIVs in wild birds and poultry combined with genetic and epidemiological data may enhance our ability to identify sources of outbreaks and LPAIV progenitors [3]. Understanding the genetic markers that facilitate evolution of LPAIV into HPAIV can increase our knowledge of AIVs with pathogenic potential and preparedness for HPAIV. Influenza A viruses carrying an intermediate DBCS could signal increased risk for mutation to HPAIV for H5 and H7 subtypes, certainly if maintained in a poultry population.

Identifying the tissue and cell types that are targeted during infection with a particular AIV strain could provide additional information, such as subtle signs of increased pathogenicity of an LPAIV leading to the emergence of HPAIV populations. The *in ovo* embryo tissue culture approach evaluated in this study may enable a rapid assessment of novel isolates and can also provide predictive value to inform disease control strategies for different strains of LPAIVs, since current approaches often apply similar statutory measures for the prevention and control of LPAIV and HPAIV [93]. The rigour of such controls reflects the perceived risk these viruses pose for mutation to HPAIV. There are economic and welfare impacts from the imposition of such measures for LPAIV infections in poultry that could be better informed if it were possible to understand and predict the mutation events with more certainty, since relatively few viruses appear to carry a high risk if they are not aggressively controlled.

## Figures and Tables

**Figure 1 viruses-12-00920-f001:**
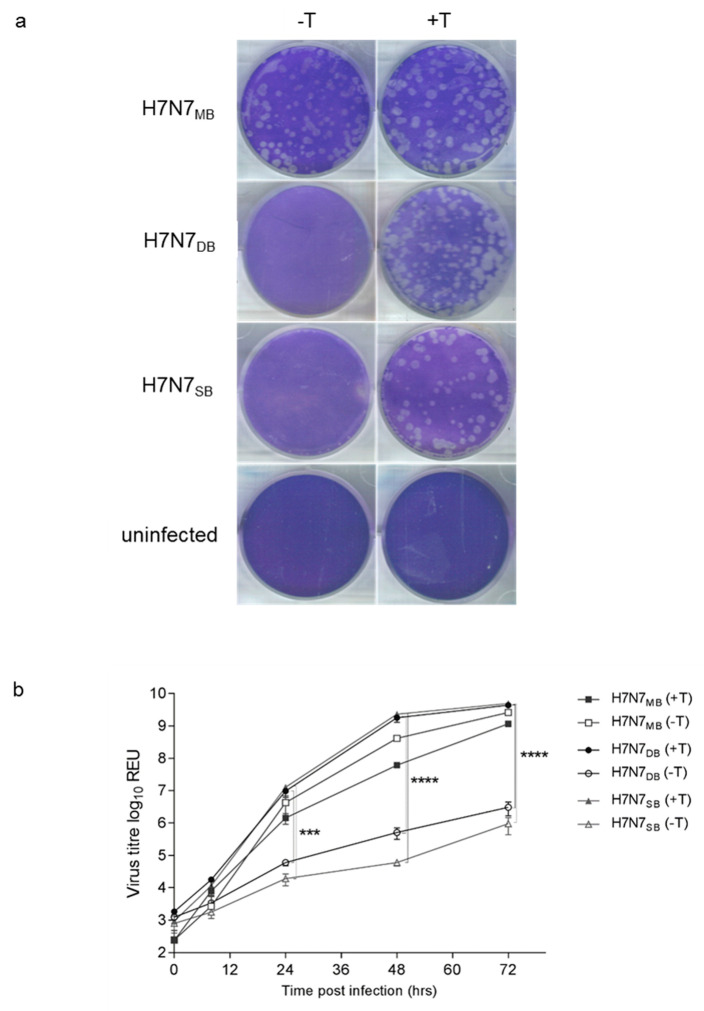
In vitro assessment of recombinant viruses. (**a**) Plaque assay using MDCK cells infected with H7N7_MB_, H7N7_DB_ and H7N7_SB_, or uninfected, in the presence (+T) or absence (−T) of TPCK trypsin (2 μg/mL). Plaque assay pictures are representative of at least two independent experiments. (**b**) Virus growth using MDCK cells at a multiplicity of infection (MOI) of 0.0001 in the presence (+T) or absence (−T) of TPCK trypsin (1 μg/mL). Viral RNA is detected by M gene RRT-PCR and measured as relative equivalent units (REU) of viral RNA against a 10-fold dilution series of median egg infectious dose (EID_50_) quantified RNA. The results are representative of two independent experiments, with standard errors indicated. ***, *p* < 0.001; ****, *p* < 0.0001.

**Figure 2 viruses-12-00920-f002:**
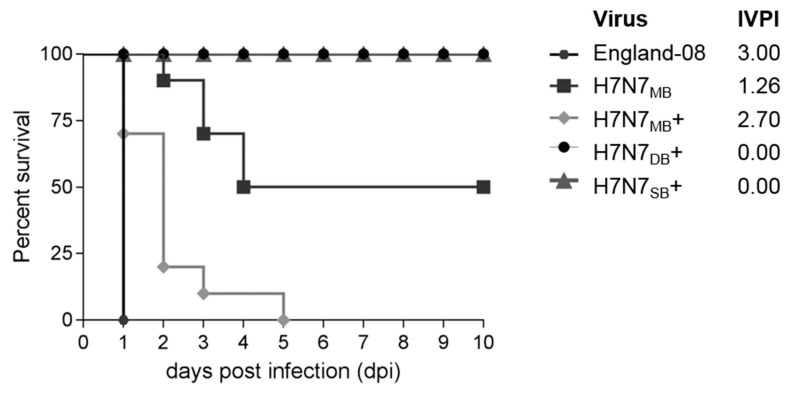
Kaplan–Meier survival curve indicating the percentage of surviving chickens inoculated intravenously. England-08 (hexagons), H7N7_MB_ (squares), H7N7_MB_+ (diamonds), H7N7_DB_+ (circles) and H7N7_SB_+ (triangles). Chickens infected with viruses containing an MBCS (England-08 [26], H7N7_MB_ and H7N7_MB_+) resulted in varying levels of mortality. Chickens infected with di-basic- (H7N7_DB_+) or single-basic- (H7N7_SB_+) containing viruses did not display any clinical signs and survived until 10 dpi when the experiment was terminated.

**Figure 3 viruses-12-00920-f003:**
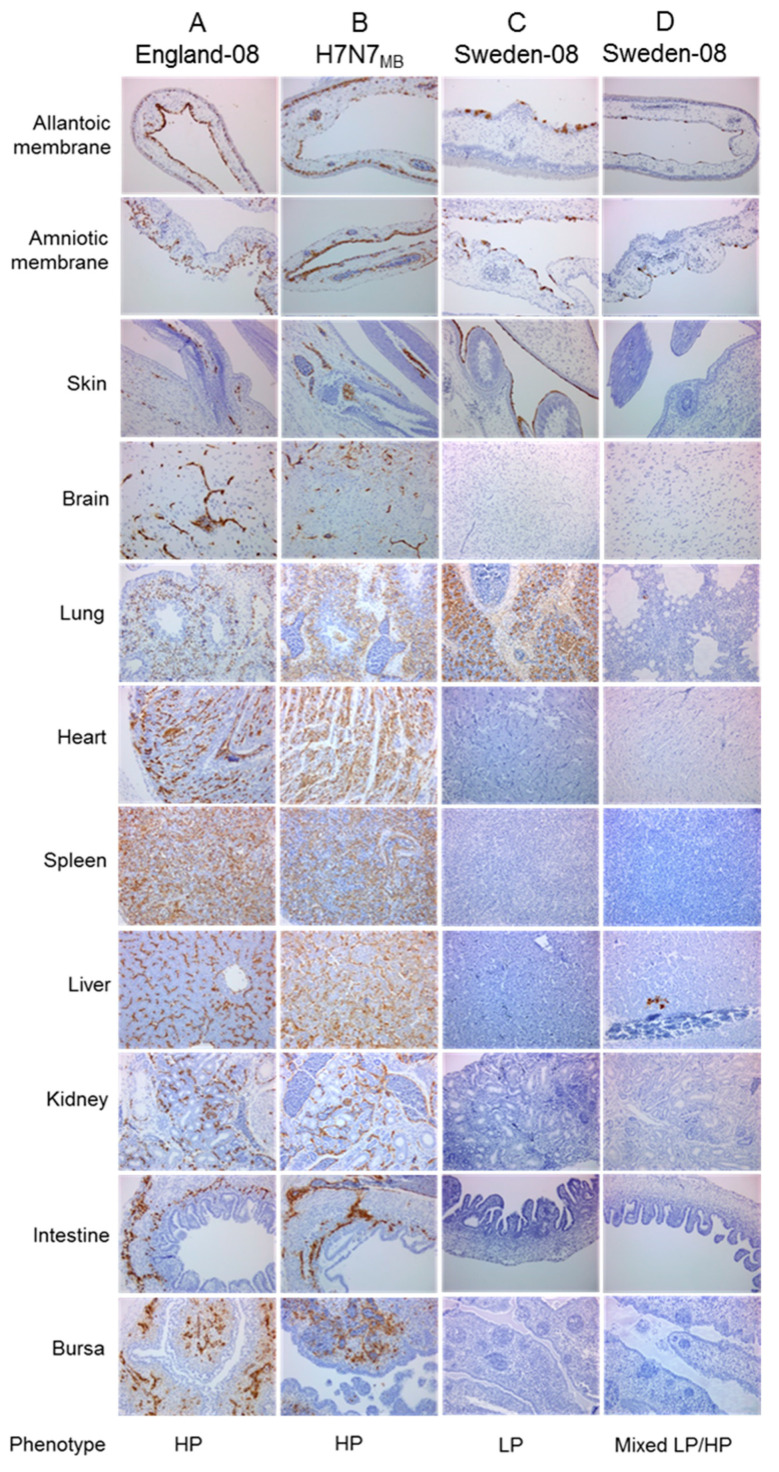
Defining H7N7 phenotypes by immunohistochemistry-stained sections of infected chicken embryos. Representative 14do EFEs infected with 10^1^ EID_50_ or 10^4^ EID_50_ England-08 (**A**) and H7N7_MB_ (**B**), and two embryos infected with 10^4^ EID_50_ Sweden-08 (**C**,**D**). Influenza A anti-NP stain shown as brown pigment. Images are at original magnification × 200. Three phenotypes were defined: (i) LP phenotype—dominant replication in the allantoic and amniotic epithelia. Virus antigen primarily detected in the allantoic epithelium with occasional spread to the amniotic epithelium. Positive cells observed in nasal, dermal and alveolar epithelial cells as well as in the air sacs and trachea/bronchi. Low numbers of immunolabelled cells observed in the liver or spleen. No consistent endothelial infection detected (**C**). (ii) HP phenotype—systemic multi-organ distribution with widespread endothelial staining. Positive labelling observed in parenchymal cells of internal organs and embryonic membranes, occasionally within neurons. Infrequently labelled dermal and nasal epithelial cells (**A**,**B**). (iii) Mixed LP/HP phenotype—replication in epithelial and dermal cells as expected for LPAIV infection but endothelial cells also labelled in multiple foci, e.g., hepatocytes as generally observed for HPAIV infection (**D**).

**Figure 4 viruses-12-00920-f004:**
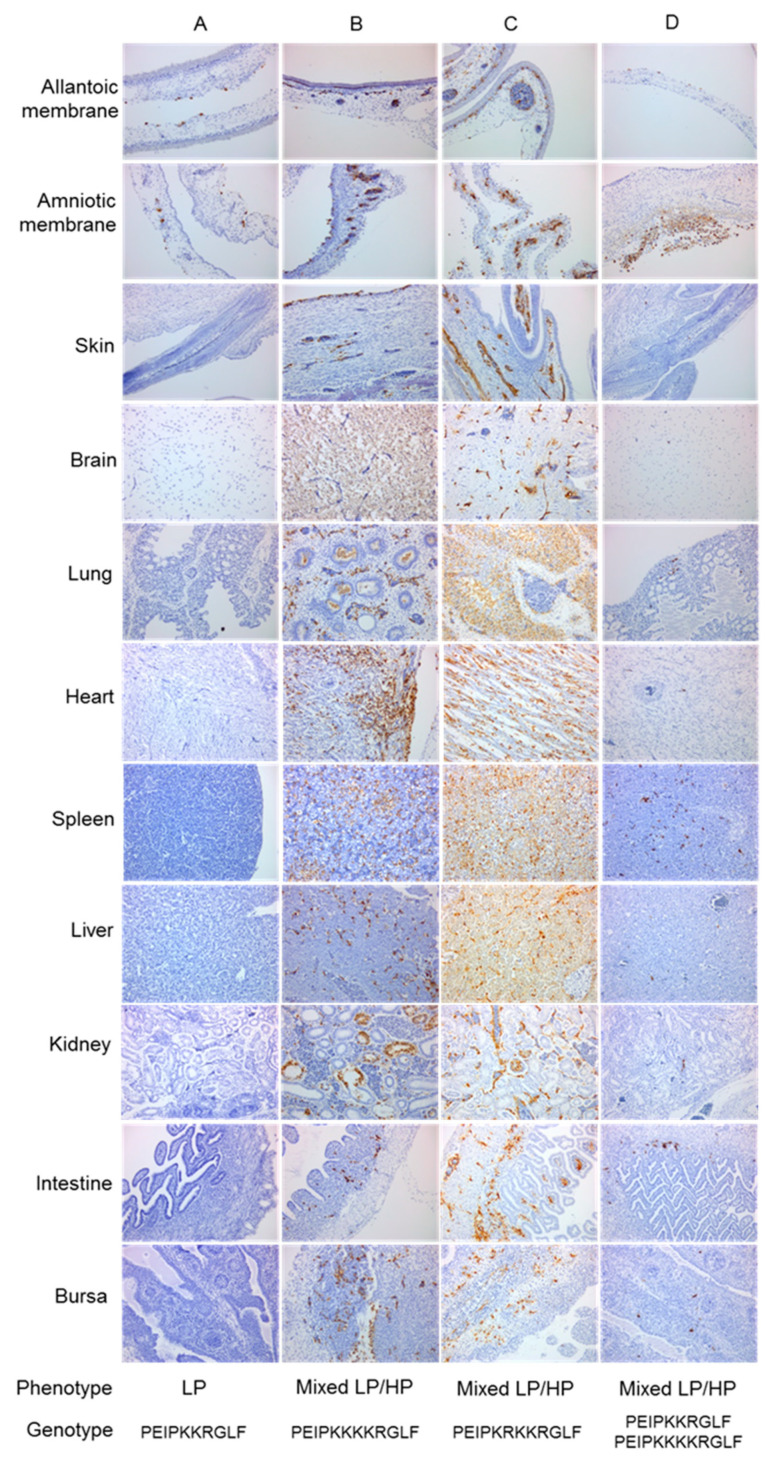
Immunohistochemistry stained sections of chicken embryos infected with H7N7_DB_ virus. Positive influenza A anti-NP stain shown as brown pigment. Images are at original magnification × 200. Recovered phenotype and genotype of each embryo are indicated. (**A**) Representative of the majority 9do and 14do embryos displaying an LP phenotype; (**B**) mixed LP/HP phenotype from a 9do embryo infected with 10^4^ EID_50_; (**C**) mixed LP/HP phenotype from a 14do embryo infected with 10^1^ EID_50_; (**D**), mixed LP/HP phenotype by IHC and MBCS by DNA Sanger sequencing from body section but LP phenotype by IHC and retention of the DBCS by DNA Sanger sequencing from the head and membrane sections of a 14do embryo infected with 10^1^ EID_50_.

**Figure 5 viruses-12-00920-f005:**
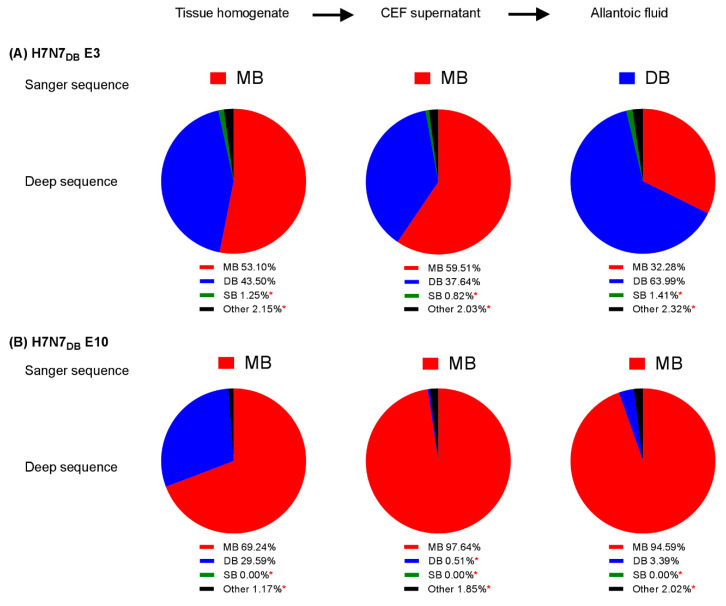
Cleavage site variant analysis from H7N7_DB_ embryo-passaged viruses. Allantoic fluid and the corresponding tissue homogenates from two embryos that mutated from DBCS to MBCS as detected by Sanger sequencing: H7N7_DB_ E3 (**A**) and H7N7_DB_ E10 (**B**). Supernatants from tissue homogenates passaged in CEFs without trypsin and allantoic fluid from EFEs inoculated with CEF supernatants were analysed for cleavage site variants and compared to results from Sanger sequencing. Variants below the error level of 1.92% (*).

**Figure 6 viruses-12-00920-f006:**
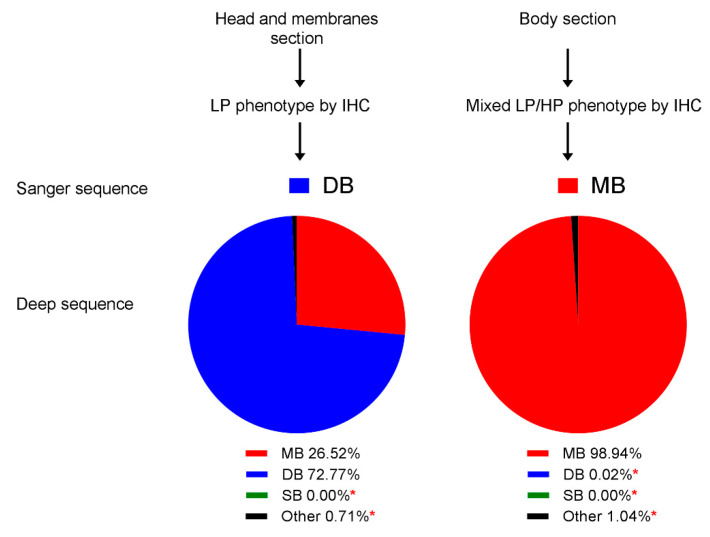
Cleavage site variant analysis from H7N7_DB_ embryo passaged virus from FFPE tissue. Variants below the error level of 1.92% (*).

**Figure 7 viruses-12-00920-f007:**
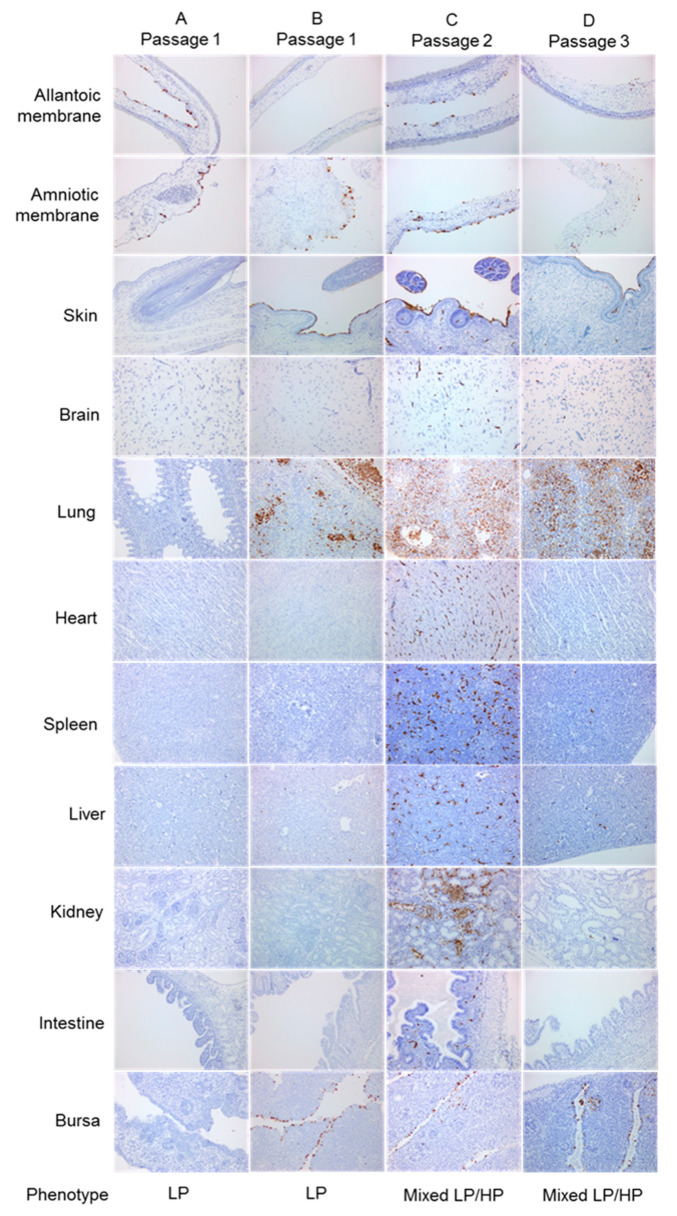
Immunohistochemistry-stained sections of chicken embryos infected with H7N7_SB_ virus. Positive influenza A anti-NP stain shown as brown pigment. Images are at original magnification × 200. (**A**) Passage 1 representative of the majority embryos displaying an LP phenotype; (**B**) passage 1 representative of an LP phenotype with enhanced labelling; (**C**) passage 2 mixed LP/HP phenotype; (**D**) passage 3 mixed LP/HP phenotype. DNA Sanger sequencing from all embryo sections showed retention of the SBCS.

**Figure 8 viruses-12-00920-f008:**
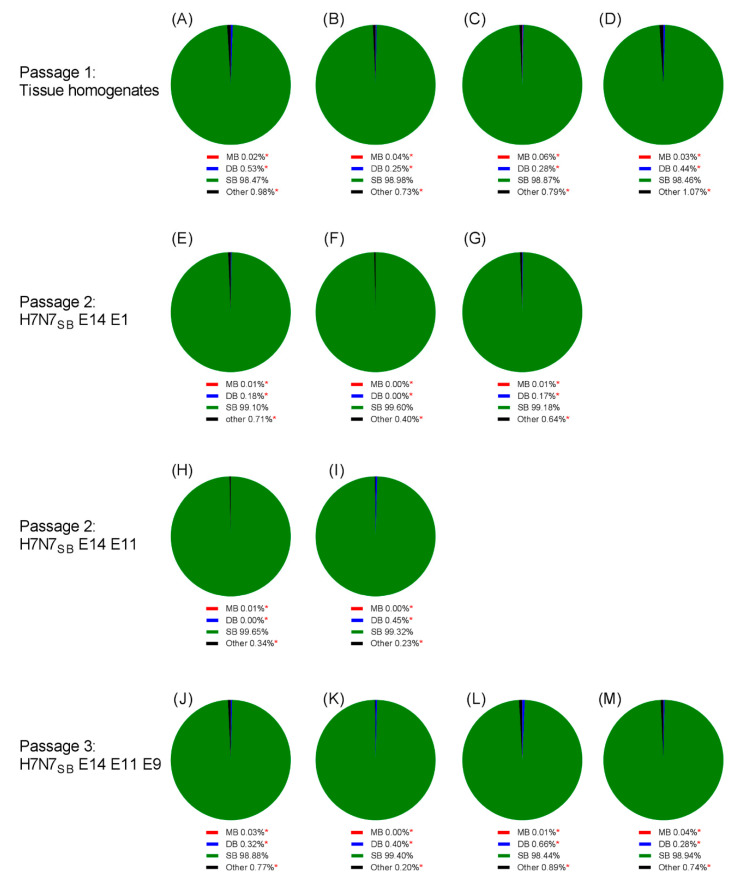
Cleavage site variant analysis from H7N7_SB_ embryo passaged viruses that were all SBCS by Sanger sequencing. Four embryo tissue homogenates that showed an enhanced level of replication in systemic tissues at passage one (**A–D**). H7N7_SB_ P2: E14, E1 tissue homogenate sequenced from PCR products (**E**) and RNA (**F**). H7N7_SB_ P2: E14, E1 FFPE tissue sequenced from PCR products (**G**). H7N7_SB_ P2: E14, E11 tissue homogenate sequenced from PCR products (**H**) and RNA (**I**). H7N7_SB_ P3: E14, E11, E9 tissue homogenate sequenced from PCR products (**J**) and RNA (**K**). H7N7_SB_ P3: E14, E11, E9 FFPE tissue sequenced from PCR products from the head and membranes (**L**) and body (**M**) sections. Variants below the error level of 1.92% (*).

**Table 1 viruses-12-00920-t001:** CS motifs for each wt and recombinant-RG virus.

Virus Name	Cleavage Site Nucleotide and Amino Acid Sequence	CS Sequence Origin
**Wildtype (wt) virus**		
England-08	**CCCGAAATCCCAAAG** AGA AAG AAA **AGA / GGCCTATTT** **P E I P K** R K K **R / G L F**	Wt H7N7 HPAIV A/chicken/England/11406/2008
Sweden-08	**CCTGAAATCCCAAAG** AAA - - - - - - **AGA / GGCCTATTT** **P E I P K** K - - **R / G L F**	Wt H7N7 LPAIV A/mallard/Sweden/100993/2008
**Recombinant-RG virus**		
H7N7_MB_ H7N7_MB_ +	**CCCGAAATCCCAAAG** AGA AAG AAA **AGA / GGCCTATTT** **P E I P K** R K K **R / G L F**	CS derived from HPAIV England-08
H7N7_DB_ H7N7_DB_ +	**CCCGAAATCCCAAAG** AAA - - - - - - **AGA / GGCCTATTT** **P E I P K** K - - **R / G L F**	Putative LP precursor virus to the HPAIV outbreak
H7N7_SB_ H7N7_SB_ +	**CCCGAAATCCCAAAG** GGA - - - - - - **AGA / GGCCTATTT** **P E I P K** G - - **R / G L F**	Typical European and Asian LP CS motif

^+^ The three different CS motifs were rescued on two H7N7 HA backgrounds. H7N7_MB_, H7N7_DB_ and H7N7_SB_ with the HA gene sequence obtained from clinical material from the outbreak, and H7N7_MB_+, H7N7_DB_+ and H7N7_SB_+ with the HA gene sequence obtained from the EFE-amplified stock, which included three nucleotide changes. Two of these nucleotide changes were non-synonymous; I77V and A125T (H7 numbering from mature protein), the latter resulting in the acquisition of a potential N-linked glycosylation site at position 123 [26].

**Table 2 viruses-12-00920-t002:** Full genome sequence differences of SBCS-passaged tissues with enhanced IHC labelling compared to A/chicken/England/11406/2008 (England-08).

	P2: E14, E1	P2: E14, E11	P3: E14, E11, E9
PB2	0	6	4
PB1	0	0	0
PA	0	8 (A20T, D27N, E59K)	24 (A20T, D27N, E59K, A85T, I100V)
HA *	0	1 (G227E)	1 (G227E)
NP	0	0	0
NA	0	1 (T51K)	1 (T51K)
M	0	0	3 (V15I)
NS	0	0	0

The number of nucleotide mutations is indicated and the non-synonymous mutations are in brackets. * Reported mutations found outside of the cleavage site, numbering from H7 open reading frame of A/chicken/England/11406/2008 (England-08).

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
