# Peer review of "The Emergence of H7N7 Highly Pathogenic Avian Influenza Virus from Low Pathogenicity Avian Influenza Virus Using an in ovo Embryo Culture Model"

_viruses, 2020, doi:10.3390/v12090920_

Round 1

Reviewer 1 Report

The manuscript entitled “The emergence of H7N7 highly pathogenic avian influenza virus from low pathogenicity avian influenza virus using an in ovo embryo culture model” investigated the mechanism of multi-basic cleavage motif acquisition of LPAI H7 virus using passages of recombinant virus containing dibasic cleavage motif in embryonic tissue, not allantoic fluid. Further, the authors found that a single basic cleavage motif conferred a limited chance to become a multi-basic cleavage motif, which is interesting. The manuscript is well-written and the authors appropriately addressed the evidence to prove their hypothesis via the current study. The authors just need to re-check the typing errors throughout the manuscript.

Reviewer 2 Report

Overview:

The authors present an examination of transition from LPAI to HPAI genotype/phenotypes and possible role of a DBCS intermediate using in ovo passage and histological analysis, with additional sequence analysis.  There are still some aspects of the study that are unclear and areas where the manuscript could be strengthened; these critiques are listed below:

General Comments:

1) There are a great deal of abbreviations used in the introduction that can be confusing at times.  Though L/HPAI as well as the SB/DB/MBCS nomenclatures are certainly necessary, it might be good to evaluate the usefulness of other lesser-used abbreviations (i.e RG and others) to clarify this section for readers not fully versed in the topic.

2) It’s not entirely clear the actual objective or hypothesis of this study.  The authors should expand upon this in lines 112-117. 

3) Figure 2 – were the H7N7DB and SB WT viruses (i.e. not the rg + viruses) tested for IVPI? and what were the values compared to the + viruses.

4) Results 3.4 – was this passage repeated in duplicate eggs per passage? Is it possible that a single passage replicate failed, but multiple replicate could have shown you the same type of low-conversion to MBCS (30%-ish) seen in the passage schemes from results 3.6? It’s not clear what exactly the experimental factor was that allowed this DB to MB transition between the two passage schemes.

5) Results 3.8 – why was only the low inoculum used (as opposed to both 10^1 and 10^4 for the DBCS).  Might a higher inoculum allowed for chance for a non-SB quasi-species to be selected for?

6) Discussion lines 447-452 – It’s a bit unclear what the authors’ conclusion is to this study.  Is it that DB represents a transition phase for further mutation? If so, can this be expanded and combined with the interesting statement at lines 511-515 about poor fitness of these DB viruses.  In general, is this DB ‘intermediate’ actually a stepping stone for more basic residue accumulation? or is there just simply an enrichment of quasi-species (likely a small population of MBCS) once it enters a gallinaceous host?  

7) Are the authors also proposing or recommending that this in ovo system can be used as a model to examine this LPAI to HPAI conversion for other experimental settings/labs?

Minor Comments:

Line 43 – suggest modifying the tense of ‘required’ to ‘require’

Reviewer 3 Report

Review report
Manuscript ID: viruses-900737
Title: The emergence of H7N7 highly pathogenic avian influenza virus from low pathogenicity avian influenza virus using an in ovo embryo culture model
Authors: Seekings et al.
Summary: Using reverse genetics, three different H7N7 viruses carrying single, double or multiple basic cleavage sites were generated. The mechanism for the shift of LP to HP pathotype was studied by passaging the first two viruses in embryonated fowl eggs. Histopathological approach was established to differentiate between LP, HP and mixed LP/HP phenotypes. Sanger and deep sequencing were performed to identify genetic changes particularly in the HA. The pathogenicity of selected viruses was assessed in chickens according to the standard IVPI.

Strength:
1. This is a compressive study used classical and advanced approaches to improve our understanding for the shift of LP H7N7 virus to HP pathotype.
2. The experiments are well designed and the manuscript is clearly written.
3. The results indicated several important aspects for the evolution of HP from LP precursors
a. LP H7N7 viruses with dibasic cleavage site may be the intermediate to generate HP viruses with multibasic cleavage site after in-ovo passaging (including 9do embryos) where the pressure is even less than in chickens. As mentioned by the authors this is due to the unstability of viruses with such dibasic cleavage site. Therefore, they are rate in nature. Secondly, results revealed that this model is useful to understand factors which may contribute to this unstability without the need for infecting chickens in compliance with the 3R concept.
b. LP H7N7 single basic cleavage site remained unchanged in this model indicating that other selection factors are required for evolution of HPAIV from viruses with single basic cleavage site.
c. Mutations beyond the cleavage site can significantly modulate virulence as conferred by the glycosylation site 123 in the HA after IVPI or spread of LPAIV H7N7 in chicken embryos after multiple passages without changes in the cleavage site. These changes might predispose or contribute in the emergence of HPAIV.

Minor comments:
1. Line 63-65: this sentence is not clear and can be deleted or examples should be added for clarification.
2. Line 71: the cited reference was on H5 viruses, please cite appropriate reference(s) for H7.
3. Line 82-83: What does it mean “single passaging method … nature”. We don’t know how many passages in birds (species!) are required for the evolution of HPAIV in nature. This sentence should be rephrased.
4. Line 115: H5 was not studied in this manuscript and should be deleted.
5. Line 140: The number of passages should be mentioned.
6. Line 178: the abbreviations “9do and 14do” should be used in the following passages.
7. Line 482: “and/or increase in interferon “ please delete or cite appropriate reference.
8. Line 505-510: Citation is required for these sentences.
9. Using IHC in this study is useful for initial screening to limit the number of samples for sequencing. However, mutations outside the cleavage site can enhance the systemic spread of the virus as seen after passaging of SBCS viruses. This is also known from other non-H5/H7 (e.g. H9N2, H4N2 or H4N6 as mentioned in the discussion section) viruses which spread systematically without mutations in the CS. The limitations of the IHC-approach should be mentioned in the discussion section.
10. How to explain that the lung tissue in figure 3 has less antigen in LP/HP mixed than in LP infected-embryos?
